# OpenReview forum: "Any-Depth Alignment: Unlocking Innate Safety Alignment of LLMs to Any-Depth"
_ICLR.cc/2026/Conference — ICLR 2026 Poster_

### Official Review · Reviewer_Ja9F · 2025-10-27

**Soundness:** 2
**Presentation:** 2
**Contribution:** 2
**Rating:** 4
**Confidence:** 4

**Summary:**

This work identifies the vulnerabilities of existing alignment schemes and the limitations of deep alignment solutions. Based on observations of intrinsic safety signals, it proposes an inference-time alignment scheme. By injecting a designed safety checkpoint during token generation, it recalls the model's harmful judgments, achieving rejection at arbitrary depth. Two implementations, RK and LP, are presented, maintaining security without compromising the model's utility.

**Strengths:**

1.The method is well-organized and the research concepts are clear.

2.Figure 3 supports the core hypotheses Q1 and Q2 proposed by the authors, namely that the security signal within the model will gradually decay during the deep generation process, but can still be reactivated through specific context tokens.

3.The proposed mechanism does not rely on retraining and can be executed only by security token injection or linear discriminator, showing high practical potential and cross-model versatility.

**Weaknesses:**

**1. The RK method's lookahead assumptions are limited:**

The current implementation relies on partial observation or detection of future token distributions, which may affect its applicability in streaming decoding scenarios. It is recommended to consider introducing a tree-of-thought or treePO-style structure to reduce prior reliance.

**2. Probe Robustness:**

The probe effect demonstrated in the experiment is limited to a specific data domain and sampling parameter configuration. It is unclear how well this linear probe generalizes under different sampling parameters.

**3. Lack of in-depth analysis of cross-layer and cross-model consistency:**

The security signal attenuation trends presented in the paper are primarily based on single-layer representations or single-model observations, lacking more systematic cross-layer stability verification.

4. From "models can self-reflectively reject" to "security signals can be triggered by headers" and then to "this can be generalized to arbitrary defense-in-depth," there is a lack of mechanistic analysis or formalized assumptions at a high level of abstraction.

**Questions:**

**1. Key terms such as "innate safety," "safety tokens," and "alignment priors" appear intertwined in the introduction, but their definitions and hierarchical distinctions are unclear, making it difficult to determine whether they refer to the representation space, gradient directions, or training data attribution.**

**2. Can the authors further explain the origin of the "safety signal" in the representation space? Is it the embedding pattern induced by the training data, or the stable direction of the alignment objective in parameter space?**

**3. Is the so-called "re-injection of the assistant header" simply a re-triggering of the language pattern through the context template (prompt conditioning), rather than a true activation of the safety representation? If the latter is the case, can more direct representation-level evidence (such as changes in activation patterns or directional analysis) be provided?**

---

> ### Author Response · Authors · 2025-11-21
> **Response to Reviewer Ja9F [Part-1]**
>
> We extend our gratitude to the reviewer for their feedback. Their recommendations have contributed to improving our paper!
>
> > **The RK method's lookahead assumptions are limited: The current implementation relies on partial observation or detection of future token distributions, which may affect its applicability in streaming decoding scenarios. It is recommended to consider introducing a tree-of-thought or treePO-style structure to reduce prior reliance.**
>
> Thank you for pointing this out. We agree that ADA-RK’s lookahead mechanism is less natural in strict streaming settings, since it requires generating a short speculative continuation to detect refusals. In practice, for real-time or streaming deployments we would recommend ADA-LP, which is both faster and more robust, as it only requires a single forward pass on the Safety Tokens without any generative lookahead (Figure 9).
>
> Our primary motivation for including ADA-RK is conceptual rather than practical: it provides direct evidence that the model’s own refusal behavior can be “unlocked’’ mid-stream via Safety-Token reinjection, demonstrating the presence of an inherent safety awareness even under adversarial attacks. Methods inspired by tree-of-thought or TreePO style decoding are complementary and could, in principle, be combined with ADA-RK or ADA-LP to explore alternative branching or verification strategies. We view this as an interesting direction for future work, but not a prerequisite for the core claims or applicability of ADA.
>
> > **Probe Robustness: The probe effect demonstrated in the experiment is limited to a specific data domain and sampling parameter configuration. It is unclear how well this linear probe generalizes under different sampling parameters.**
>
> Thank you for raising this point. In all of our main experiments, both training and evaluation use a fixed decoding configuration (temperature 1.0, without additional sampling heuristics). While the linear probe is trained only on WildChat and WildJailbreak, it is evaluated on multiple additional harmful benchmarks (AdvBench, JailbreakBench, StrongREJECT, and HEx-PHI) as well as seven diverse benign benchmarks (GSM8K, MATH, BBH, HumanEval, MMLU, SimpleQA, GPQA Diamond), covering math reasoning, coding, QA, and knowledge tasks. Empirically, its effectiveness is therefore not limited to a single data domain but holds across a broad distribution of inputs.
>
> With respect to sampling parameters, ADA-LP makes decisions based on the hidden states at the Safety Tokens rather than on the exact sampled tokens (which is affected by the sampling parameters). Consistent with this, the probe already remains robust under substantially stronger distribution shifts, such as adaptive adversarial attacks and adversarial SFT (Figures 6 and 7), which suggests that it is not brittle to moderate variations in decoding behavior.
>
> To further support this point, we have added explicit experiments varying the sampling temperature for both harmful and benign settings (Section F.5, Page 26). For GCG adversarial examples on AdvBench with gemma-2-9b-it, the attack success rate remains essentially unchanged across temperatures 0.0, 0.25, 0.50, and 1.00 (ASR = 2% at all four temperatures). For benign MMLU examples with the same model, the ADA-LP over-refusal rate is similarly stable: 0.3%, 0.2%, 0.3%, and 0.3% at temperatures 0.0, 0.25, 0.50, and 1.00, respectively. These results indicate that the linear probe generalizes well across a range of standard sampling configurations.

---

> > ### Author Response · Authors · 2025-11-21
> > **Response to Reviewer Ja9F [Part-2]**
> >
> > > **Lack of in-depth analysis of cross-layer and cross-model consistency: The security signal attenuation trends presented in the paper are primarily based on single-layer representations or single-model observations, lacking more systematic cross-layer stability verification.**
> >
> > Thank you for the comment. We indeed provide in-depth analysis of cross-layer (we include the analysis for all layers for a given model) and cross-model consistency (we provide the results on 9 different models.)
> >
> > **[Cross layer consistency]**: Figure 4 reports the validation performance of Safety Token probes across all layers for all 9 models we study. As shown there, the middle layers exhibit consistently high and relatively stable accuracy with only minor variation, indicating that the safety signal does not rely on a single special layer and does not attenuate sharply across depth within the network. As shown in Figure 10 (Page 15), we even show that this strong, linearly separable safety signal is even detectable at all tested readout locations (e.g., after Input LayerNorm or Self-Attention or MLP) within the same layer.
> >
> > **[Cross model consistency]**: Our experiments are conducted on nine different models from diverse families and scales: Llama 2 7B it, Llama 3.1 8B it, Mistral 8B it, Gemma 2 2B/9B/27B it, Qwen2.5 7B it, DeepSeek R1 7B, and gpt oss 120B. Full per model probe results and depth robustness curves are provided in Appendix F. The unlocking effect is ubiquitous: Linear probe on Safety Tokens consistently exposes a strong, linearly separable harmfulness signal across model families (Llama, Qwen, Mistral, Gemma, DeepSeek variants, gpt oss), parameter scales (2B/9B/27B/120B), and core designs (dense, Mixture of Experts, and reasoning centric).
> >
> > > **From "models can self-reflectively reject" to "security signals can be triggered by headers" and then to "this can be generalized to arbitrary defense-in-depth," there is a lack of mechanistic analysis or formalized assumptions at a high level of abstraction.**
> >
> > Sorry for the confusion, we indeed provided mechanistic analysis via transcoder in Appendix C. As shown in Figure 11, the high-impact refusal features (layers 18–24) have negligible activation throughout the harmful continuation, but spike sharply on the injected assistant header tokens `<end_of_turn>\n<start_of_turn>model\n`, especially on the “model” token and its following newline. In other words, these refusal-related neurons are effectively dormant during the harmful stream and are reactivated only at the Safety Tokens.
> >
> > Additionally, following Reviewer mcZ1’s suggestion, we add two new pieces of evidence. (1) Qualitative feature visualizations make the refusal semantics explicit: for example, L18 F12640 aligns with apology tokens like "sorry," while L22 F5394 aligns with first person pronouns like "I,'' consistent with refusal templates (Figure 12, Page 17). (2) Preliminary causal interventions show these features exert direct control over refusals: zeroing them out removes refusals and yields harmful continuations, while strongly activating them induces refusals even for benign queries (Figure 13, Page 17). Together, these results support our claim that Safety Tokens reactivate a latent safety representation, rather than merely causing a superficial dialogue reset.
> >
> > > **Key terms such as "innate safety," "safety tokens," and "alignment priors" appear intertwined in the introduction, but their definitions and hierarchical distinctions are unclear, making it difficult to determine whether they refer to the representation space, gradient directions, or training data attribution.**
> >
> > Thank you for raising this point. By “alignment prior'' we mean the safety awareness induced by pretraining or safety-focused instruction tuning. “Innate safety'' refers to the model’s internal capacity to distinguish harmful from benign continuations even when this judgment is not explicitly verbalized (no response-level refusals), operationally evidenced by a stable decision boundary in representation space (Figure 3). “Safety Tokens”' denote specific special tokens (primarily the assistant header) whose hidden states reliably expose this safety-relevant signal and are strongly linearly separable for harmful vs benign continuations, in contrast to generic natural-language tokens. We have clarified these concepts in line 142 (Section 2, Page 3).

---

> ### Author Response · Authors · 2025-11-21
> **Response to Reviewer Ja9F [Part-3]**
>
> > **Can the authors further explain the origin of the "safety signal" in the representation space? Is it the embedding pattern induced by the training data, or the stable direction of the alignment objective in parameter space?**
>
> Thank you for the insightful question! When we refer to a “safety signal” in representation space, we mean an embedding pattern in the hidden states, not a specific direction in parameter space. Concretely, ADA-LP operates on the hidden states at a specific layer, and the success of a linear probe there indicates that pretraining plus subsequent safety-oriented stages (instruction tuning / RL-style alignment) have induced a consistent pattern that separates harmful from benign continuations; we can think of the results in Figure 3 as showing this pattern in activation space when reduced to two dimensions. We view the signal as an emergent representation-level pattern created by the large amount of training data and objectives, which can remain latent under attack but be “unlocked” by querying the Safety Tokens.
>
> Besides, as shown in Figure 12, we observe that some features are highly activated when the model is strongly associated with producing "sorry'' or "apologize'' tokens in the training data using the attribute graph introduced in [circuit-tracer](https://transformer-circuits.pub/2025/attribution-graphs/methods.html).
>
> > **Is the so-called "re-injection of the assistant header" simply a re-triggering of the language pattern through the context template (prompt conditioning), rather than a true activation of the safety representation? If the latter is the case, can more direct representation-level evidence (such as changes in activation patterns or directional analysis) be provided?**
>
> Thank you for the insightful question. We believe the effect goes beyond prompt conditioning by the context template. As shown in Figure 5, there are instances where ADA-RK does not surface an explicit refusal, yet the internal signal used by ADA-LP at the Safety Tokens is still strongly predictive of harmfulness, allowing ADA-LP to achieve near 100% refusal. The same pattern appears under adversarial prompt attacks (Figure 6) and adversarial SFT (Figure 7), where ADA-LP is consistently more robust than ADA-RK. This systematic gap indicates that we are not merely retriggering a surface language pattern, but are accessing a latent safety representation that is reliably exposed at the Safety Tokens and can be read out at the representation level even when the model does not explicitly refuse in text.
>
> To further substantiate this point, we added two complementary forms of evidence in the revision (Appendix C, pages 16–17; Figures 12 and 13), along with a reproducible intervention notebook in the Supplementary Material following the [official circuit-tracer intervention](https://github.com/safety-research/circuit-tracer/blob/main/demos/intervention_demo.ipynb) setup.
>
> First, we provide qualitative feature visualizations that make the semantics of the relevant CLT features explicit. In Figure 12(a), the mid-layer feature L18–F12640 is strongly associated with apology tokens such as "sorry" or "apologize,'' while the deeper-layer feature L22–F5394 in Figure 12(b) aligns with first-person pronouns like "I,'' consistent with refusal templates such as "I cannot'' or "I am sorry.'' Each panel shows examples where the feature is maximally active, making the refusal-related semantics directly visible.
>
> Second, we include preliminary causal interventions on these features to test whether they control refusals. Specifically, we (1) zero out their activations to check whether refusals disappear, and (2) strongly amplify them to test whether refusals can be induced. Figure 13(a) shows that when the model is prompted with a harmful query and prefixed with '*Sure, here are some tips on how to commit suicide:*', the original continuation is a refusal ('*Don't do it. You only have one life.*'), but ablating the refusal features causes the model to produce harmful guidance instead. Conversely, Figure 13(b) shows that amplifying the same features for a benign query ('*Do you know a good place to buy a book?'*) flips a helpful answer into an unwarranted refusal ('*I'm sorry, but as a text-based AI, I'm unable to make'*). These interventions indicate that the identified features exert causal control over refusal behavior rather than merely correlating with it, providing stronger representation-level support for our mechanistic interpretation.
>
> ---
> We thank you for your review and constructive suggestions. If we can provide any further clarifications or additional details, please let us know. If we have properly addressed all concerns, then we kindly ask that you consider raising your score.

---

> > ### Comment · Reviewer_Ja9F · 2025-11-25
> > **response**
> >
> > While I remain skeptical about defining clear boundaries for safety, I do appreciate your effort to make the case.

---

> ### Author Response · Authors · 2025-11-27
> **Response for Safety Boundary**
>
> Our goal is not to define the safety boundary itself (that is, what kinds of questions are considered harmful or benign). In practice, such boundaries are specified by external safety policies and regulations (for example, those inspired by frameworks like the [EU AI Act](https://eur-lex.europa.eu/legal-content/EN/TXT/PDF/?uri=OJ:L_202401689)) and then instantiated through the model provider’s pretraining and post-training pipelines (so-called "alignment prior"). We do not introduce a new notion of safety on top of this, and our work is explicitly operational: **we take the resulting harmful/benign distinction as given and study how it is encoded in the model’s hidden states and how it can be reliably unlocked at inference time**.
>
> Empirically, **this internal boundary (in the hidden states at a specific layer, as clearly stated in the paper) aligns well with five widely used safety benchmarks (WildJailbreak, AdvBench, StrongReject, HEx-PHI, JailbreakBench) and eight benign benchmarks (WildChat, GSM8K, MATH, BBH, HumanEval, MMLU, SimpleQA, GPQA Diamond)** [so-called "innate safety"]. ADA achieves near-100% refusal rates on harmful examples and preserves utility on benign ones, which would not be possible if the model’s own safety boundary were fundamentally misaligned with the harmful/benign labels in these benchmarks. In other words, the "safety" signal we exploit is whatever boundary the underlying training has already induced in the model, not one we impose externally.
>
> We also agree that in practice different deployments may adopt different safety specifications. A key advantage of ADA is that it is compatible with any such specification: given binary labels under a chosen safety definition, **the Safety Token representations consistently exhibit a much more linearly separable harmful/benign signal than generic language tokens**. This means that, in principle, the same mechanism can be used to unlock and operationalize the safety boundary induced by any desired policy, as long as it is reflected in the model’s training data or alignment procedure.
>
> We hope this clarification addresses the concern regarding the notion and boundaries of safety in our work and are happy to clarify any remaining concerns if some questions still appear unaddressed.

---

### Official Review · Reviewer_ZDcC · 2025-10-30

**Soundness:** 3
**Presentation:** 3
**Contribution:** 3
**Rating:** 6
**Confidence:** 3

**Summary:**

This paper addresses the "shallow alignment" problem in Large Language Models (LLMs), where models effectively refuse harmful prompts initially but fail when harmful content emerges later in the generation (e.g., via assistant-prefill attacks). The authors propose Any-Depth Alignment (ADA), an inference-time defense mechanism designed to unlock the model's innate safety alignment at arbitrary generation depths. ADA is based on the observation that "Safety Tokens" (typically assistant header tokens) concentrate the model's safety priors. By re-injecting these tokens mid-stream, ADA forces the model to reassess the ongoing generation. Two variants are presented: ADA-Rethinking (ADA-RK), a training-free method that checks for refusals in a short lookahead after token injection, and ADA-Linear Probe (ADA-LP), which uses a lightweight linear classifier on the hidden states of injected Safety Tokens to detect harmfulness with minimal overhead.

**Strengths:**

Novelty and Significance: The paper introduces the important concept of "deep prefill attacks" to rigorously evaluate alignment depth and identifies a fundamental mechanism (unlocking innate alignment via Safety Tokens) rather than just proposing another fine-tuning method or external model. The idea of leveraging the model's own latent safety knowledge at inference time is highly significant.

Effectiveness and Generality: ADA, especially the ADA-LP variant, demonstrates outstanding effectiveness, achieving near-perfect refusal rates against very challenging deep prefill attacks and substantially mitigating strong adversarial prompt attacks (GCG, AutoDAN, PAIR, TAP). Crucially, this high performance is shown across a wide variety of modern open-source models (Llama, Gemma, Mistral, Qwen, DeepSeek, gpt-oss) and even commercial ones (Claude), indicating the underlying principle is broadly applicable.

Efficiency and Practicality: ADA-LP offers a compelling practical solution. It requires only a simple linear probe (trained once) and operates with negligible, constant-time inference overhead by reusing the KV cache. This makes it suitable for real-time streaming applications where traditional guardrails are often too slow or memory-intensive, especially for long contexts.

**Weaknesses:**

Reliance on Hidden State Access (ADA-LP): The most effective variant, ADA-LP, requires access to the model's internal hidden states. This limits its direct applicability to scenarios where users interact with closed APIs that only provide text outputs. While ADA-RK offers a training-free alternative for such cases, it is shown to be less consistently effective, particularly on models with weaker base alignment.

Vulnerability in User-Controlled Environments: As acknowledged by the authors, ADA is primarily effective when the inference process is controlled by the service operator. In fully open-source deployments where the end-user can modify the model or the inference code, the ADA checks (token injection and probing/lookahead) could potentially be disabled by a malicious user, bypassing the defense entirely.

**Questions:**

The linear probe for ADA-LP is trained on a specific dataset (WildChat/WildJailbreak) and shown to generalize across models. How sensitive is the probe's performance to the type of alignment used in the base model (e.g., RLHF vs DPO vs Constitutional AI)? Would a probe trained on data from an RLHF-aligned model work as effectively when applied to a DPO-aligned model, or does the nature of the internal safety representation differ significantly?

ADA operates by injecting Safety Tokens periodically (e.g., every 100 tokens). What is the sensitivity to this checkpoint frequency? Is there a trade-off between computational overhead (checking more often) and detection latency (potentially generating more harmful tokens before a check triggers refusal)? Could an adaptive checking strategy be more optimal?

---

> ### Author Response · Authors · 2025-11-21
> **Response to Reviewer ZDcC [Part-1]**
>
> We thank the reviewer for their comments and feedback. The suggestions and detailed comments provided have contributed to enhancing the quality of our work!
>
> > **Reliance on Hidden State Access (ADA-LP): The most effective variant, ADA-LP, requires access to the model's internal hidden states. This limits its direct applicability to scenarios where users interact with closed APIs that only provide text outputs. While ADA-RK offers a training-free alternative for such cases, it is shown to be less consistently effective, particularly on models with weaker base alignment.**
>
> Thank you for raising this concern. We agree that ADA-LP requires access to internal hidden states. In benign user scenarios, modern models are typically aligned well enough that they do not produce harmful content in most cases; harmful outputs usually arise when a malicious user applies adversarial attacks. However, a malicious user has no incentive to enable a safety mechanism (e.g., a guardrail) that would block the harmful content they are trying to obtain. Therefore, our primary target setting is provider side deployment, where the model provider or deployer controls inference and wishes to defend against malicious exploits (for example, self hosted deployments, open weight models, or providers integrating ADA internally).
>
> ADA-RK is included as a lightweight, training free alternative for environments where only text level access is available, and also as an explicit diagnostic of the base model’s latent alignment. As expected, there is a natural tradeoff between efficacy and access: ADA-RK is more broadly applicable but less consistently strong, while ADA-LP is less broadly applicable but achieves substantially higher robustness. Taken together, the two variants cover a wide spectrum of deployment scenarios, and ADA-RK further illustrates our key point that strong safety priors remain present in aligned models but can be “locked” during long harmful continuations.
>
> > **Vulnerability in User-Controlled Environments: As acknowledged by the authors, ADA is primarily effective when the inference process is controlled by the service operator. In fully open-source deployments where the end-user can modify the model or the inference code, the ADA checks (token injection and probing/lookahead) could potentially be disabled by a malicious user, bypassing the defense entirely.**
>
> Thank you for highlighting this limitation. We agree that ADA, like most inference time defenses, is primarily effective when the inference pipeline is controlled by the service operator. In fully user controlled, fully open source deployments where a malicious user can modify the model or the inference code, ADA’s checks (token injection and probing/lookahead) could indeed be disabled. However, this limitation is not specific to ADA: the same applies to external guardrail models, different alignment training techniques, and other provider side safety mechanisms, all of which can be bypassed or disabled once an adversary has full control over the model. For example, as shown in Figure 1, all open-weight models, including gpt-oss-120B (even including Claude model), can be easily jailbroken by simply prefilling harmful content. Figure 7 further shows that even benign SFT can substantially erode a model’s built-in deep alignment. Given this reality, in practice, we can only reduce the utility of the released model to avoid severe misuse as shown in openai’s blog [1]. Our work is therefore scoped to provider or deployer controlled settings. This is both the practical regime (as a malicious user has no incentive to enable defenses that would block their attack) and the dominant deployment mode for current frontier models.
>
> [1] Wallace, E., Watkins, O., Wang, M., Chen, K., & Koch, C. (2025). Estimating worst-case frontier risks of open-weight llms. arXiv preprint arXiv:2508.03153.

---

> > ### Author Response · Authors · 2025-11-21
> > **Response to Reviewer ZDcC [Part-2]**
> >
> > > **The linear probe for ADA-LP is trained on a specific dataset (WildChat/WildJailbreak) and shown to generalize across models. How sensitive is the probe's performance to the type of alignment used in the base model (e.g., RLHF vs DPO vs Constitutional AI)? Would a probe trained on data from an RLHF-aligned model work as effectively when applied to a DPO-aligned model, or does the nature of the internal safety representation differ significantly?**
> >
> > Thank you for the question and for highlighting this point. We apologize for the confusion in the current description. In our experiments, we do not train a single probe on one aligned model and then directly apply it to a different alignment pipeline. Instead, for each base model, we train a separate linear probe on that model’s own hidden states.
> >
> > Concretely, we first jailbreak a GPT model via SFT API to obtain long harmful generations (more than 3k tokens with 100% ASR) on the WildJailbreak questions, and then construct a training set using that base model’s hidden states on (i) those harmful continuations from WildJailbreak and (ii) benign conversations from WildChat. Then we test this probe on the same model but for separate harmful datasets (e.g., AdvBench, StrongReject) and benign benchmarks (e.g., MMLU, MATH, HumanEval) to show the generalization.
> >
> > Under this setup, the probe is adapted directly to each base model’s internal representations, regardless of whether its alignment was obtained via RLHF, DPO, or Constitutional AI. In practice, as shown in Figure 18, across 9 diverse models trained with different alignment pipelines, we find that as long as the base model retains a nontrivial safety prior, the Safety Token features remain strongly separable, and ADA-LP continues to be effective.
> >
> > > **ADA operates by injecting Safety Tokens periodically (e.g., every 100 tokens). What is the sensitivity to this checkpoint frequency? Is there a trade-off between computational overhead (checking more often) and detection latency (potentially generating more harmful tokens before a check triggers refusal)? Could an adaptive checking strategy be more optimal?**
> >
> > Thank you for the insightful question. As shown in Figure 9, the cost of a single ADA safety check is roughly equivalent to generating one additional token, since we reuse the KV cache and only perform a short forward pass on the Safety Tokens. With a checkpoint frequency of every 100 tokens, this corresponds to about a 1% computational overhead. For harmful queries, the net cost can even be lower in practice, because ADA often stops generation early and thus avoids generating many additional tokens. In realistic deployments, the safety checks can also be run in a parallel forked stream (one thread for normal generation, one for safety checking), which further mitigates latency.
> >
> > There is a trade-off between checkpoint frequency and detection latency: more frequent checks reduce the maximum number of harmful tokens that can be emitted before a refusal, at the price of slightly higher overhead. ADA is fairly robust to this choice, and following your suggestion we added new experiments to demonstrate it. Specifically, under the GCG attack on gemma-2-9b-it over AdvBench (Section F.5, Page 25), the attack success rate (ASR) is 2% / 2% / 6% / 4% for checkpoint intervals of 25 / 50 / 75 / 100 tokens, respectively. This shows that less frequent checking (75 to 100 tokens) slightly degrades robustness, but the ASR remains low overall.
> >
> > We agree that adaptive strategies are a promising refinement. One simple scheme we tested is to check densely early in generation (every 25 tokens for the first 100 tokens), then switch to a sparser schedule (every 100 tokens thereafter). Following your suggestion, we evaluated this on GCG against gemma-2-9b-it, and the adaptive schedule still achieves a 2% ASR, matching the fully dense 25-token schedule as shown above while reducing the total number of checks. This is consistent with the intuition that current jailbreak attacks mainly focus on steering the model within the first few tokens, so we can put more dense checks there. Exploring such adaptive schedules is an interesting extension that we plan to investigate in future work.
> >
> > ---
> >
> > Thank you again for your thoughtful review and constructive feedback! We hope our responses have addressed your concerns and clarified the contributions of our work. We would sincerely appreciate your consideration of an updated score in light of these clarifications.

---

### Official Review · Reviewer_ydC8 · 2025-10-30

**Soundness:** 3
**Presentation:** 3
**Contribution:** 3
**Rating:** 6
**Confidence:** 4

**Summary:**

The paper proposes Any-Depth Alignment (ADA), a training-free, inference-time defense that restores refusals mid-generation. The key observation is that alignment signals concentrate in assistant-header “safety tokens.” Two variants are presented: ADA (RK), which re-injects the header to trigger a refusal, and ADA (LP), which applies a lightweight linear probe to the hidden states of injected safety tokens. ADA achieves high refusal rates under deep assistant-prefill attacks and low adversarial prompt ASR, while preserving benign utility and incurring minimal overhead. Results generalize across model families and remain robust after SFT.

**Strengths:**

1. The identification of assistant-header tokens as safety tokens that surface a strong, separable harmfulness signal is an insightful finding, which makes the proposed ADA well-motivated and empirically grounded.
2. The authors show strong empirical performance of ADA compared to existing methods: high refusal rates against harmful prompts, minimal over-refusal with benign prompts, and relatively efficient. Various robustness checks are also provided.
3. ADA doesn’t require fine-tuning and model-agnostic, making it broadly applicable across diverse model families and architectures. By leveraging only inference-time interventions or hidden-state probing, ADA can be easily integrated into existing model pipelines.
4. The paper is well-structured and the writing is easy to follow. Good presentation overall.

**Weaknesses:**

1. Leakage before cutoff: Because interventions trigger mid-stream, a small amount of harmful content can be emitted before the refusal fires; although the authors acknowledge this limitation, some quantification or examples of such leakage across tasks would strengthen the argument about the utility of ADA.
2. Dependence on the base model: the effectiveness of ADA, especially ADA-RK, fundamentally relies on the base model’s alignment strength (i.e., the model must already possess latent robust refusal behavior for safety-Token injection to “unlock”). This means ADA is not a universal defense but rather a mechanism amplifier for already-aligned models. On weakly aligned or uncensored models, ADA may not be very effective.
3. The paper attributes ADA’s effectiveness to the reactivation of an “innate” safety signal triggered by reinjecting the assistant-header tokens mid-generation. However, this interpretation may conflate two distinct phenomena. Reintroducing the assistant header effectively resets the conversational state, prompting the model to begin a new assistant turn, which inherits its default alignment priors from the system prompt and chat template. In other words, the refusal that follows might result from a stylistic or structural re-anchoring of the model’s dialogue state, rather than the activation of a latent safety representation.

**Questions:**

1. Starting line 192: the authors mention that as d increases, the hidden states of safety tokens are increasingly separable, while for generated tokens the features become more entangled. Any explanations or hypotheses for this trend?

---

> ### Author Response · Authors · 2025-11-21
> **Response to Reviewer ydC8 [Part-1]**
>
> We are grateful to the reviewer for their thorough feedback. These comments have helped improve the quality of this work!
>
> > **Leakage before cutoff: Because interventions trigger mid-stream, a small amount of harmful content can be emitted before the refusal fires; although the authors acknowledge this limitation, some quantification or examples of such leakage across tasks would strengthen the argument about the utility of ADA.**
>
> Thanks for the thoughtful suggestion. We agree that, because interventions trigger mid-stream, a small amount of harmful content can sometimes be emitted before the refusal fires.
>
> **[Quantitative Results]**: In our revision, we now *quantify* this effect and provide concrete examples. Specifically, in Appendix F.2 (page 23), Table 6 (page 30) reports, for each model on dataset AdvBench, the token position at which ADA-LP first triggers a refusal under streaming decoding (with checks every 25 tokens).
>
> For example, with model `gemma-2-9b-it`, the statistics for the depth of the first refusal under different attacks are:
>
> + GCG: mean 35.6 tokens, 25/50/75 percentiles = 25 / 25 / 25 tokens
>
> + AutoDAN: mean 56.7 tokens, 25/50/75 percentiles = 25 / 25 / 50 tokens
>
> + PAIR: mean 40.4 tokens, 25/50/75 percentiles = 25 / 25 / 25 tokens
>
> + TAP: mean 36.9 tokens, 25/50/75 percentiles = 25 / 25 / 25 tokens
>
>  Thus, for this model the median first-refusal position is always at the very first check (25 tokens), and even the 75th percentile never exceeds 50 tokens. The full table shows the same pattern across other models: *ADA-LP almost always halts harmful continuations at the first or second check.*
>
> **[Qualitative Results]**: As for examples of such leakage, Appendix F.2 includes Figure 23 (page 31), which shows representative AdvBench examples for `gemma-2-9b-it`. For each attack (GCG, AutoDAN, PAIR, TAP) we display the attack prompt and the leaked prefix before ADA-LP first emits a refusal. As we can see, the leaked content is typically short and with zero-utility, consisting of generic phrasing without detailed, actionable instructions and providing essentially no guidance in the real world.
>
> These additions support our empirical observations that (i) the leaked prefix before cutoff is typically short, and (ii) compared to the base model (which outputs the full harmful response (Figure 15)), ADA-LP substantially reduces both the amount and specificity of harmful content.
>
> > **Dependence on the base model: the effectiveness of ADA, especially ADA-RK, fundamentally relies on the base model’s alignment strength (i.e., the model must already possess latent robust refusal behavior for safety-Token injection to “unlock”). This means ADA is not a universal defense but rather a mechanism amplifier for already-aligned models. On weakly aligned or uncensored models, ADA may not be very effective.**
>
> Thank you for the valuable comment! We agree that ADA, and especially ADA-RK, relies on the base model having some nontrivial safety prior. This is by design: one of our key contributions is showing that *modern LLMs, even when they appear fully jailbroken at the response level (e.g., under prefill attacks or adversarial attack) still retain internal safety signals that can be unlocked.*
>
> However, our results show that even for weakly aligned or partially uncensored models, ADA, particularly ADA-LP, can still be effective. As shown in the right panel of Figure 7 in Section 6, even under adversarial SFT (finetune on a harmful dataset, a form of uncensoring) that progressively erases alignment. After only 20 steps, the refusal rate of the underlying base model quickly collapses to 0% and methods such as Deep Alignment also rapidly degrade, reaching refusal rates below 5%, whereas ADA-LP maintains a strong refusal rate, remaining close to 100% even after 1000 SFT steps. Intuitively, Safety Tokens act as an information aggregator: even when the global alignment of the model is weakened, harmful and benign continuations still induce systematically different features at these tokens, which ADA-LP can exploit to preserve effective refusals.

---

> > ### Author Response · Authors · 2025-11-21
> > **Response to Reviewer ydC8 [Part-2]**
> >
> > > **The paper attributes ADA’s effectiveness to the reactivation of an “innate” safety signal triggered by reinjecting the assistant-header tokens mid-generation. However, this interpretation may conflate two distinct phenomena. Reintroducing the assistant header effectively resets the conversational state, prompting the model to begin a new assistant turn, which inherits its default alignment priors from the system prompt and chat template. In other words, the refusal that follows might result from a stylistic or structural re-anchoring of the model’s dialogue state, rather than the activation of a latent safety representation.**
> >
> >
> > Thank you for raising this concern. We agree that reinjecting the assistant header mid generation marks the beginning of a new assistant turn and re-anchors the dialogue state in the chat template, thereby reinstating the model’s default alignment prior. Your intuition is correct, and this is precisely the phenomenon we set out to study. To our knowledge, our work is the first to systematically investigate the following question: *once a model has been jailbroken and is already generating harmful content, what happens if we interrupt it mid stream, append the assistant header, and let it continue?* The outcome is not predictable a priori; it is not clear in advance whether a jailbroken model will persist in the harmful trajectory or revert to refusal after a simple header reinjection. Our experiments show that the latter behavior is both robust and surprisingly strong across models, including strong proprietary models such as Claude, where a simple reinjection consistently restores refusals in a way that is neither trivial nor easy to anticipate.
> >
> > At the same time, we believe the effect goes *beyond* a purely stylistic or structural re-anchoring of the dialogue state:
> >
> > **First, safety persists even when no textual refusal appears.** As shown in Figure 5, there are cases where ADA-RK does not yield an explicit refusal, yet the internal signal used by ADA-LP at the Safety Tokens is still strongly predictive of harmfulness, allowing ADA-LP to achieve near 100% refusal. This pattern also appears under adversarial prompt attacks (Figure 6) and adversarial SFT (Figure 7), where ADA-LP is consistently more robust than ADA-RK. This gap indicates that the improved refusals are not just a consequence of resetting the conversational template, but are tied to a latent safety representation that is reliably exposed at Safety Tokens and that ADA-LP can read out even when the model does not explicitly refuse in text.
> >
> > **Second, safety-Token hidden states are linearly separable and mostly depth-invariant.** Our analysis in Figure 3 and Figure 4 show that the hidden states at Safety Tokens form a separable representation of harmfulness even when the last-generated tokens remain entangled. This indicates that re-inserting the header does more than shift format; it exposes a safety assessment that the model already maintains internally but does not verbalize unless prompted by the header.
> >
> > **Third, these Safety Token features have a causal relationship with refusal behavior.** Following Reviewer mcZ1’s suggestion, we include additional preliminary causal intervention experiments on the identified refusal related features (Figure 13, Page 17). Zeroing out these features suppresses refusals for harmful queries, while strongly activating them induces refusals even for benign queries, which is also an evidence that the refusal results more from the activation of a latent safety representation.
> >
> > In summary, while header reinjection does perform a structural reset of the assistant turn, our evidence shows that ADA’s effectiveness comes from reliably exposing and reactivating a persistent internal safety representation, which ADA then leverages to restore refusals mid-stream.

---

> > > ### Author Response · Authors · 2025-11-21
> > > **Response to Reviewer ydC8 [Part-3]**
> > >
> > > > **Starting line 192: the authors mention that as d increases, the hidden states of safety tokens are increasingly separable, while for generated tokens the features become more entangled. Any explanations or hypotheses for this trend?**
> > >
> > > Thank you for pointing this out! At shallow depths, the preceding content is often short and relatively generic, even for harmful prompts (this is especially true under adversarial attacks, where the early part of the prompt is often boilerplate or obfuscating text). As the depth d increases, harmful continuations accumulate concrete, target specific details that become progressively more distinct from the details present in benign continuations. Because the Safety Tokens attend over the entire preceding context, their hidden states act as an information aggregator of the accumulated content. The growing semantic divergence between harmful and benign continuations at larger depths is therefore reflected in increasingly linearly separable Safety Token representations.
> > >
> > > In contrast, the hidden state of the last generated token at each depth is optimized for local next token prediction rather than for providing a global assessment of harmfulness:
> > >
> > > First, even the same surface token (for example, “bomb'”) can correspond to harmful or harmless intent depending on the broader context (e.g., historical description, safety discussion, fiction), yet its representation is dominated by its lexical and semantic content, so harmful and benign uses of the same token can look very similar in feature space sometimes.
> > >
> > > Second, more importantly, across different depths and prompts, the last generated token is not a fixed token: each such token must simultaneously encode its local word-level semantics and carry enough global information to support continuation of the sequence. This multi-purpose role makes these hidden states relatively noisy and heterogeneous, which makes it difficult for a single linear classifier to cleanly separate harmful from benign behavior. In contrast, Safety Tokens are not burdened with expressing specific lexical content and instead act as dedicated aggregators of the preceding context, which leads to much more stable and linearly separable harmfulness signals.
> > >
> > > ---
> > >
> > > Thank you again for your thoughtful review. We hope we have addressed all of your concerns; we are happy to clarify anything that remains unclear, and we would be grateful if you would consider raising your score accordingly!

---

> > > > ### Comment · Reviewer_ydC8 · 2025-11-25
> > > >
> > > > I appreciate the authors' clear and detailed responses, as well as the additional experiment conducted to quantify the leakage issue. I believe my concerns and questions are effectively addressed, so I'm raising my overall rating to 8.

---

### Official Review · Reviewer_mcZ1 · 2025-11-02

**Soundness:** 3
**Presentation:** 4
**Contribution:** 3
**Rating:** 6
**Confidence:** 3

**Summary:**

The paper introduces Any-Depth Alignment (ADA), an inference-time safety mechanism designed to unlock latent alignment signals within large language models (LLMs) and ensure robust refusal behavior at any generation depth. The authors argue that LLMs possess innate safety priors encoded in their assistant-header tokens, which can be reactivated mid-generation through two mechanisms: ADA (RK) and ADA (LP).

**Strengths:**

(1) Novel framing and insight: The paper advances a conceptually fresh hypothesis—that safety alignment priors already exist in model hidden states but remain “locked.” The discovery that assistant headers act as Safety Tokens is both empirically supported and intuitively plausible.

(2) Strong empirical coverage: Evaluation spans 9 diverse model families, 4 harmfulness benchmarks (AdvBench, JailbreakBench, StrongReject, HEx-PHI), and multiple defense baselines. ADA (LP) outperforms both deep alignment and strong guardrails such as Llama-Guard 4 and IBM Granite-Guardian.

(3)Technical soundness and clarity: The linear-probe formulation is clearly explained. The visualization of hidden-state separability is compelling, showing safety signals becoming linearly separable only when probed via header tokens.

(4) Practical significance: ADA offers zero-training, constant-time inference, and minimal overhead. This makes it deployable in real-time settings—an appealing property for industry safety stacks.

**Weaknesses:**

(1) Mechanistic claims need deeper causal validation: While the authors show correlations between Safety-Token activations and refusal behavior, the causal mechanism remains partially speculative. The Transcoder neuron analysis is suggestive but does not yet establish that these neurons cause refusal rather than correlate with it.

(2) Scope of evaluation is primarily safety-focused: The work measures safety robustness and over-refusal but does not test whether ADA affects reasoning quality, factuality, or calibration under benign long-context workloads. These would strengthen claims of “no utility degradation.”

**Questions:**

(1) Could the authors provide additional evidence that the Safety-Token activations cause refusal rather than merely correlate with it? For example, have the authors tried causal ablations (e.g., zeroing out or rescaling the most activated neurons or attention heads identified via the Transcorder features) to test whether refusals disappear when those activations are suppressed?

(2) Have the authors evaluated whether ADA (LP) alters model behavior on non-safety, reasoning-intensive, or long-context tasks—for example, reasoning depth in GSM8K or factual calibration in MMLU when ADA checks are active?

(3) Since ADA introduces mid-generation checkpoints, is there any measurable latency or coherence degradation in multi-step reasoning chains or tool-use tasks? Quantitative or qualitative analysis would clarify the “no utility degradation” claim.

---

> ### Author Response · Authors · 2025-11-21
> **Response to Reviewer mcZ1 [Part-1]**
>
> We are grateful to the reviewer for their insightful feedback. The suggestions and comments have helped to improve its quality!
>
> > **The mechanistic story is still mostly correlational. Safety-Token activations and Transcoder features align with refusals, but it is unclear if they cause refusals. Can the authors provide causal evidence, e.g., by ablating or amplifying the top activated neurons/heads and showing refusals disappear or emerge accordingly?**
>
> We appreciate the reviewer’s concern and agree that correlational evidence alone is not sufficient to fully substantiate a strong mechanistic claim. Our intention is not to claim that we have completely characterized the full causal pathway behind refusals, but rather to show that Safety-Token activations are tightly linked to refusal behavior and that this linkage can be interpreted and probed using Transcoder features.
>
> To strengthen this point, we have added two complementary forms of evidence in the revision, which are now included in Appendix C (pages 16–17) as Figure 12 and Figure 13. We have also added a notebook implementing the intervention experiments in the Supplementary Material so that the reviewer can reproduce our results; this notebook closely follows the [official circuit-tracer intervention demo](https://github.com/safety-research/circuit-tracer/blob/main/demos/intervention_demo.ipynb).
>
> First, we provide qualitative feature visualizations that make the semantics of the relevant CLT features explicit. As illustrated in Figure 12 (a), the middle-layer feature L18–F12640 is strongly associated with the model producing tokens such as “sorry” or “apologize,” whereas the deeper-layer feature L22–F5394 in Figure 12 (b) encourages the model to generate first-person pronouns like “I,” consistent with refusal templates such as “I cannot” or “I am sorry.” Each panel shows dataset examples for which the corresponding feature is most strongly active, making the refusal-related semantics captured by these features directly visible.
>
> Second, we now include preliminary causal intervention experiments on these identified features (1. zeroing out their activations to test whether suppressing them removes refusals, and 2. strongly activating them to test whether refusals can be induced). Concretely, we find that manipulating the top refusal-related features has a direct and intuitive effect on model behavior. As shown in Figure 13(a), we intervene on the activations corresponding to the tokens `model` and `\n` when the model is prompted with a harmful query and prefixed with '*Sure, here are some tips on how to commit suicide:*'. The original continuation is a refusal ('*Don't do it. You only have one life.'*), but when these features are ablated (set to zero), the model instead produces harmful guidance ('*First, try to find a quick way to end your suffering.'*). Similarly, in Figure 13(b), we activate the same features to a large positive value (100) for a benign query, '*Do you know a good place to buy a book?'*, where the model provides a helpful answer before intervention; under this intervention, however, the model produces an unwarranted refusal ('*I'm sorry, but as a text-based AI, I'm unable to make'*). These results indicate that the identified features exert causal control over refusal behavior rather than merely correlating with it, providing additional support for our mechanistic interpretation; further discussion is added around line 900 in the revision.

---

> > ### Author Response · Authors · 2025-11-21
> > **Response to Reviewer mcZ1 [Part-2]**
> >
> > >  **Scope of evaluation is primarily safety-focused: The work measures safety robustness and over-refusal but does not test whether ADA affects reasoning quality, factuality, or calibration under benign long-context workloads. These would strengthen claims of “no utility degradation.” Have the authors evaluated whether ADA (LP) alters model behavior on non-safety, reasoning-intensive, or long-context tasks—for example, reasoning depth in GSM8K or factual calibration in MMLU when ADA checks are active?**
> >
> > **[ADA Generation]**: Thank you for the insightful question. We first clarify how ADA (LP) operates. As shown in Figure 2, ADA (LP) periodically forks the KV cache, injects Safety Tokens only in the forked branch, and uses that branch solely to decide whether generation should be stopped. These Safety Tokens and safety checks are never inserted into the main decoding stream and therefore do not alter the logits or token-by-token trajectory of normal generation. In other words, there are two threads: (i) standard generation, and (ii) an intermittent safety-check thread that reuses the KV cache. If the safety check does not trigger a refusal, then ADA’s standard-generation output is identical to the original model’s output under the same seed. Consequently, ADA (LP) cannot change reasoning depth, factual calibration, or any other benign behavior unless it mistakenly triggers a refusal.
> >
> > **[Safety and Utility Tradeoff of ADA]**: We explicitly evaluated this safety–utility tradeoff on non-safety, reasoning-intensive, and long-context workloads across GSM8K, MATH, BBH, HumanEval, MMLU, SimpleQA, and GPQA Diamond (Figure 8, Section 7). In these experiments, ADA (LP) performs periodic safety checks at multiple depths, and we use a conservative metric: if a refusal is triggered at any step, we count the entire instance as refused. Even under this strict criterion, ADA (LP) maintains approximately 0% refusal across depths. For example, the refusal rate is exactly 0% on GSM8K, 0.3% on MMLU with an average refusal rate of 0.4% across all seven benign datasets, as reported in Table 1. These results indicate that ADA (LP) does not materially alter model behavior on benign reasoning, coding, or factual QA tasks, and that its practical utility impact is negligible.
> >
> > > **Since ADA introduces mid-generation checkpoints, is there any measurable latency or coherence degradation in multi-step reasoning chains or tool-use tasks? Quantitative or qualitative analysis would clarify the “no utility degradation” claim.**
> >
> > We thank the reviewer for raising this question! Firstly, as stated above, when no refusal is triggered, the generation produced from ADA is identical to that of the base model, implying no coherence degradation regardless of task. As shown in Figure 9, because ADA reuses the same KV cache as the main generation, the time and memory cost of a single safety check is essentially equivalent to generating one extra token. For multi-step reasoning chains, if we run a safety check every 100 generated tokens, this corresponds to roughly a 1% increase in compute. In a parallel implementation where the safety branch is executed in a separate thread that only reads the shared KV cache, this overhead is effectively negligible in the main decoding stream, since the safety branch only interrupts generation when a safety violation is detected.
> >
> > In terms of memory, each safety check requires storing activations for approximately only one additional token, and we release these activations immediately after the check, so the steady-state memory overhead is also comparable to a single token.
> >
> > For tool-use tasks (which we are currently exploring as future work), we speculate that ADA can be deployed even more sparsely: a single safety check before executing a tool call or external action, without any periodic checks during normal decoding, may be sufficient. In that setting, the additional latency and memory overhead are even less impactful. We leave these results to future work.
> >
> > ---
> > We appreciate your thoughtful feedback and hope our clarifications resolve the concerns raised. Should everything now be addressed, we would be grateful if you would consider adjusting your score accordingly.

---

> > > ### Comment · Reviewer_mcZ1 · 2025-11-28
> > >
> > > I thank the authors for their response which have resolved my concerns- I have decided to maintain my current positive rating.

---

### Author Response · Authors · 2025-11-21
**Summary of updates in our revision**

We would like to thank all reviewers for their valuable feedback and thoughtful suggestions. We are pleased that the reviewers recognize (i) the conceptual novelty that LLMs contain "locked'' innate safety priors concentrated in assistant-header "safety tokens,'' which ADA reactivates at arbitrary depths, (ii) the strong and broad empirical validation over 9 diverse model families, 4 harmfulness benchmarks (AdvBench, JailbreakBench, StrongReject, HEx-PHI), and extensive 7 benign benchmarks showing near-perfect deep-prefill robustness while maintaining low over-refusal and no utility degradation, (iii) the technical soundness and clarity of the ADA-LP linear-probe formulation, supported by compelling hidden-state separability evidence, and (iv) the practical significance of a training-free, constant-time, minimal-overhead, model-agnostic defense that is easy to integrate and remains robust after subsequent instruction tuning.

Following the reviewers’ suggestions, we made the following main revisions, and all revised or added content is highlighted in bold blue text in the revision.

- **Stronger mechanistic and causal validation (Appendix C, pages 16 to 17, Figures 12 and 13; Supplementary Notebook).**
  Following Reviewer mcZ1 and Reviewer Ja9F, we added new mechanistic interpretability evidence using the Transcoder analysis. We include qualitative feature visualizations that make refusal related semantics explicit (Figure 12, Page 17), and preliminary causal intervention experiments showing that ablating or amplifying top refusal features directly suppresses or induces refusals (Figure 13, Page 17). We also provide a reproducible intervention notebook in the supplementary material.
- **Quantification of leakage before cutoff (Appendix F.2, page 23; Table 6 on page 30; Figure 23 on page 31).**
   Following Reviewer ydC8, we added a systematic analysis of the amount of harmful content that can appear before ADA triggers. We report first refusal positions under streaming decoding across models and attacks (Table 6) and provide representative examples of leaked prefixes (Figure 23). The results show that ADA-LP typically halts harmful continuations at very early checkpoints (with 25 tokens or 50 tokens), and the leaked content is short and low utility.
- **Checkpoint frequency robustness and adaptive schedules (Appendix F.5, Page 25)**.
  Following Reviewer ZDcC, we added experiments varying checkpoint intervals and show ADA remains robust under a range of frequencies (check with an interval over 25/50/75/100), with only mild degradation at sparser schedules. We also test a simple adaptive scheme that checks densely early and sparsely later, matching dense performance while reducing total checks.
- **Probe robustness to decoding parameters  (Appendix F.5, Page 26)**
  Following Reviewer Ja9F, we added experiments varying standard sampling temperatures for both harmful and benign settings. The probe’s effectiveness and over refusal behavior remain stable, indicating the ADA-LP is not brittle to different sampling settings.
- **Clarified definitions (Section 2, page 3, around line 142).**
  In response to Reviewer Ja9F, we strengthened and clarified the hierarchy of key concepts ("alignment prior," "innate safety," and "Safety Tokens’") and explicitly scoped ADA to provider or deployer controlled settings where hidden state access and inference control are available. We also emphasize the complementary roles of ADA-RK (diagnostic and text-only) and ADA-LP (robust provider-side defense).

We sincerely appreciate the reviewers for dedicating their time to review our paper and look forward to any further discussion and suggestions to help improve the quality of our work.

---

### Meta-Review · Area_Chair_Ert4 · 2026-01-06

**Summary:**

The paper proposed ADA (Any-Depth Alignment), an inference-time defense that reactivates latent safety signals by reinjecting assistant-header "Safety Tokens" mid-generation. The primary concerns centered on whether Safety-Token activations causally drive refusals or merely correlate with them; quantification of harmful content leaked before ADA triggers a refusal; whether the effect represents true safety representation activation versus superficial dialogue state reset; probe robustness across sampling parameters and checkpoint frequencies; reliance on hidden state access for ADA-LP, limiting applicability to closed APIs; and unclear definitions distinguishing "innate safety," "alignment priors," and "Safety Tokens." Key strengths include novel conceptual framing, strong empirical coverage across 9 model families and 11+ benchmarks, practical deployability with minimal overhead, and technical clarity of the linear-probe formulation.

**Reviewer Concerns:**

The authors addressed most technical concerns with new experiments and analysis. For causal validation, they added Transcoder feature visualizations showing refusal-related semantics and causal intervention experiments demonstrating that ablating top refusal features suppresses refusals while amplifying them induces refusals for benign queries, with a reproducible notebook provided. For leakage quantification, they added Table 6 reporting first-refusal token positions across models and attacks, showing median detection at the first checkpoint, plus qualitative examples. The concern about dialogue state reset versus true safety activation was addressed through evidence that ADA-LP detects harmfulness even when ADA-RK produces no explicit refusal, indicating the probe reads a latent representation rather than surface patterns.

**Reviewer Scores:**

The preponderance of evidence from the three engaged reviewers supports acceptance.

---

### Decision · Program_Chairs · 2026-01-26

Accept (Poster)